# Effects of Different Conditions on Co-Pyrolysis Behavior of Corn Stover and Polypropylene

**DOI:** 10.3390/polym12040973

**Published:** 2020-04-22

**Authors:** Fengze Wu, Haoxi Ben, Yunyi Yang, Hang Jia, Rui Wang, Guangting Han

**Affiliations:** 1Key Laboratory of Energy Thermal Conversion and Control of Ministry of Education, Southeast University, Nanjing 210096, China; 18827421336@163.com (F.W.); 15251933285@163.com (Y.Y.); 18707126420@163.com (H.J.); wangrui@163.com (R.W.); 2State Key Laboratory of Bio-Fibers and Eco-Textiles, Qingdao University, Qingdao 266071, China; hanguangting@163.com

**Keywords:** plastic, biomass, co-pyrolysis, pyrolysis oils, nuclear magnetic resonance analysis

## Abstract

The pyrolysis behavior of corn stover and polypropylene during co-pyrolysis was studied using a tube furnace reactor. The effects of mixing ratio of corn stover and polypropylene, pyrolysis temperature, addition amount of catalyst (HZSM-5) and reaction atmosphere (N_2_ and CO_2_) on the properties of pyrolysis products were studied. The results showed that co-pyrolysis of corn stover and polypropylene can increase the yield of pyrolysis oil. When corn stover:polypropylene = 1:3, the yield of pyrolysis oil was as high as 52.1%, which was 4.5% higher than the theoretical value. With the increase of pyrolysis temperature, the yield of pyrolysis oil increased first and then decreased, and reached the optimal yield at 550 °C. The addition of catalyst (HZSM-5) reduced the proportion of oxygenates and promoted the generation of aromatic hydrocarbons. CO_2_ has a certain oxidation effect on the components of pyrolysis oil, which promoted the increase of oxygen-containing aromatics and the reduction of deoxy-aromatic hydrocarbons. This study identified the theoretical basis for the comprehensive utilization of plastic and biomass energy.

## 1. Introduction

In recent years, the rapid development of industry, the large consumption of fossil energy and the demand for environmental protection have promoted the research and development of renewable energy. Abundant, renewable, uncompetitive with food and easily available, biomass is a good alternative to fossil fuels [1,2] which has attracted increasing attention from researchers at home and abroad. Among all thermochemical conversions, rapid pyrolysis is considered as one of the most promising methods due to its high liquid yield, including thermal degradation of waste in anaerobic or low-oxygen environments, which can control the formation of harmful substances such as dioxins [3,4,5]. 

Although bio-oils are environmentally friendly, due to the characteristics of the biomass feedstock, the bio-oils prepared by pyrolysis have high acidity and viscosity and their fuel characteristics are still lower than fossil fuels, especially in terms of combustion efficiency [6]. In this case, the high content of oxygenates in the pyrolysis oil [7,8,9,10] is the main reason limiting its further industrial application. While plastic is rich in hydrogen, traditional landfills are not only difficult to decompose, but also waste land resources, as burning not only wastes energy, but also produces dioxins to pollute the environment [11]. In addition, the co-pyrolysis technology can significantly improve the quantity and quality of pyrolysis oil without any catalyst or high-pressure hydrogenation [1]. Therefore, feeding hydrogen-rich plastics with oxygen-rich biomass is a simple, cheap and effective way to obtain advanced pyrolysis oil [12,13,14]. The co-pyrolysis of biomass and plastics is considered to be a promising technology that can improve the quality of oil, which also paves the way for a better utilization of municipal solid waste plastics [15]. 

At present, due to the diversity of experimental methods and biomass materials, there are various experiments on co-pyrolysis of biomass and plastics, and the conclusions drawn are also different. Özsin et al. [16] studied the co-pyrolysis behavior of polystyrene (PS), polyethylene terephthalate (PET), polyvinyl chloride (PVC) and walnut shell, peach pit at 500 °C. The results showed that the co-pyrolysis of PET, PS and biomass could effectively improve the liquid yield, but PVC inhibited the formation of bio-oil. Pinto et al. [17] studied the effect of co-pyrolysis of cellulose and PS at different mass ratios on the composition of liquid products. The results revealed that the mass ratio of Cellulose/PS would greatly affect the contents of oxygenated compounds and hydrocarbons in bio-oils. To our knowledge, there is currently no study on the analysis of biomass-plastic co-pyrolysis oil by nuclear magnetic resonance (NMR). Due to the short reaction time, the chain length of alkenes and alkanes produced by the depolymerization of chain plastics may be as high as C_40+_ during the rapid pyrolysis process [18]. In order to analyze the physicochemical properties of pyrolysis oil more accurately and to resolve the limitations of gas chromatography–mass spectrometry (GC–MS), this study used NMR technology to perform the co-pyrolysis of corn stover(CS) and polypropylene(PP). This is a more effective method to analyze the composition of pyrolysis oil [19,20,21,22].

This study explored the effects of material mixing ratio, reaction temperature, catalyst addition amount, and reaction atmosphere on the co-pyrolysis product yield and the physicochemical properties of pyrolysis oil. In addition, there are few studies on the reaction process and synergistic effect during the co-pyrolysis of biomass and plastic. Therefore, the interaction mechanism of CS and PP co-pyrolysis and modification of pyrolysis oil were also discussed in this study, which provided new ideas and a theoretical basis for the comprehensive utilization of plastic and biomass resources.

## 2. Materials and Methods 

### 2.1. Material Preparation

CS was collected on a farm in Lianyungang, Jiangsu Province, China. The obtained CS was ground and sieved with a 60-mesh sieve, and dried in a vacuum drying box at 60 °C for 48 h before pyrolysis. PP powder (~100 mesh) was purchased from HuaChuang Plastic Co., Dongguan, China. Table 1 summarized the proximate and ultimate analyses of CS and PP. ZSM-5 (SiO_2_/Al_2_O_3_ = 46) was purchased from Nankai University Catalyst Co., Ltd., Tianjin, China. Prior to the experiments, the ZSM-5 zeolite was converted to HZSM-5 by calcining in nitrogen at 600 °C for 6 h.

### 2.2. Tube Furnace Pyrolysis

In this study, a tube furnace device was employed to pyrolyze the raw materials, which was introduced in our team’s previous research [23]. One side of the device is a gas supply unit, which can be connected to nitrogen and carbon dioxide gas. Prior to heating, control the mass flow controller to ensure that the gas flow rate is 150 mL/min and hold for 30 minutes to exhaust air from the system. Then, the heating time and reaction temperature of the tube furnace were adjusted. When the box reached the required temperature, immediately put the quartz tube containing the raw materials into the box to heat and set the reaction time to 15 minutes. The oil from the reaction flowed through the condenser equipped with −10 °C ethanol to the collection bottle and was washed out with dichloromethane after the reaction for subsequent chemical analysis. The gas entered the gas sampling bag through the dry tube and was discharged after treatment. The oil and carbon yields were calculated by an electronic balance, and the gas yields were calculated by the difference.

### 2.3. Elemental Analysis of Pyrolysis Oil

The constituents of pyrolysis oil was explored by using the EURO EA3000 element analyzer (EuroVector Inc., Pavia, Italy). The hydrogen and carbon contents were obtained by combustion and the oxygen content was confirmed by mass difference.

### 2.4. NMR Analysis of Pyrolysis Oil

All NMR samples were derived from 100 mg pyrolysis oils dissolved in 1000 µL deuterated chloroform (CDCl_3_-d_1_). AVANCE III HD 600MHz NMR spectrometer (Bruker Inc., Zurich, Switzerland) was used to obtain ^13^C NMR spectra of pyrolysis oil generated under different conditions, with operating parameters set to 1500 scans and 3s pulse delay [19,22]. For ^1^H NMR, the operation parameters were set to 16 transients and 5 s pulse delay. All NMR spectra were processed by MestReNova v12.0 (Mestrelab Research, Santiago De Compostela, Spain).

### 2.5. Synergy Analysis

In order to study the interaction between CS and PP co-pyrolysis, the synergistic effects of product yields and the synergistic effects of physicochemical properties of pyrolysis oil were evaluated, respectively. Based on the weight loss of pyrolysis of CS and PP alone, the theoretical yield of pyrolysis products was calculated using Equation (1). The difference between the experimental and theoretical yield was expressed by Equation (2). Based on the physicochemical properties of CS and PP separately pyrolyzed oils, Equation (3) was used to calculate the theoretical content of the physiochemical properties of the co-pyrolysis oil. The difference between the experimental and theoretical contents of physiochemical properties is expressed by Equation (4).
Ycal = Xcs Ycs + Xpp Ypp,(1)
ΔY = Yex − Ycal,(2)
Zcal = (Xcs Ycs Zcs + Xpp Ypp Zpp)/(XcsYcs + XppYpp),(3)
ΔZ = Zex − Zcal,(4)
where Xcs and Xpp represent the mass percentages of CS and PP in the feedstock, respectively. Ycs and Ypp represent the experimental yields of CS and PP separately pyrolyzed, respectively. Zcs and Zpp represent the physicochemical properties of CS and PP separately pyrolyzed oil, respectively. Ycal and Yex represent the calculation and experimental yield of co-pyrolysis, respectively. Zcal and Zex represent the calculation and experimental content of physiochemical properties of co-pyrolysis oil, respectively.

## 3. Results and Discussion

### 3.1. Co-Pyrolysis Analysis of CS and PP in Different Proportions

#### 3.1.1. Product Yield Analysis

The yield of products pyrolyzed at 500 °C with different CS/PP ratios (1:0, 3:1, 1:1, 1:3, 0:1) is shown in Figure 1. At 500 °C, the oil production rate of PP alone pyrolysis reached 57.1%, far higher than that of CS pyrolysis alone, which was 19.1%. Compared with the higher oil yield, the carbon generated by PP pyrolysis alone was only 0.2%, as PP is rich in volatiles and relatively low in fixed carbon and ash contents. With the increase of PP content in the co-pyrolysis, it is worth noting that the yield of pyrolysis oil increased from 30.8% to 52.1%. In addition, compared with CS pyrolysis alone, the addition of PP can significantly reduce the carbon yield. When the CS/PP ratio was 1:3, the carbon yield was at a minimum of 6.8%. On the one hand, the carbon produced by PP pyrolysis alone was lower than that of CS pyrolysis alone. On the other hand, because the H/C_eff_ of PP is higher than CS, When the content of PP increases, PP acts as a hydrogen donor and combines with the unstable free radicals generated during pyrolysis of CS, which inhibits the polymerization and cross-linking reactions, leading to the decrease of carbon yield [24,25].

#### 3.1.2. Synergy Analysis of Product Yield

The experimental and calculated yields of different CS/PP ratios pyrolyzed at 500 °C are summarized in Table 2. When CS:PP = 1:1 was co-pyrolysis at 500 °C, the calculated yields of pyrolysis oil, carbon and gas were 38.1%, 14.9% and 47.0%, respectively. It can be seen from Table 2 that the carbon and gas yields were both lower than the theoretical values, while the liquid yields were higher than the theoretical values. This is due to the positive synergistic effect of CS and PP co-pyrolysis on the yield of pyrolysis oil, which increases the liquid yield. Moreover, with the increase of PP content in the co-pyrolysis, the synergistic effect became more pronounced. Because pyrolysis occurs through free radicals [26], CS pyrolysis generates a large number of small free radicals, while PP pyrolysis generates relatively large free radicals. When CS and PP are co-pyrolyzed, the small free radicals generated by CS cross-react with large free radicals generated by PP to promote the formation of high-molecular weight organic compound oils [27], thus inhibiting the formation of low-molecular weight gas compounds.

#### 3.1.3. Elemental Analysis of Pyrolysis Oil

The C, H, and O analysis of pyrolysis oil produced from different CS/PP ratios at 500 °C co-pyrolysis is shown in Table 3. As can be seen from the Table 3, when CS was pyrolyzed alone, the oxygen content in pyrolysis oil was as high as 29.6 wt %, while the deoxidation rate of bio-oil was only 40.9%. When CS:PP = 1:1 was co-pyrolysis, the generated oils contained a large amount of C and H, while the content of O was only 6.3 wt %, and the deoxidation rate of pyrolysis oil was up to 74.9%. From the equation of Lloyd and Davenport [28]: HHV (MJ/kg) = –0.3578C – 1.1357H + 0.0845O − 0.0594N − 0.1119S, where C, H, O, N and S are mass percentages of carbon, hydrogen, oxygen, nitrogen and sulfur, respectively. The higher heating value (HHV) of the bio-oil pyrolyzed by CS alone was 28.44 MJ/kg. The low HHV limits its widespread use as a renewable energy source. The HHV of oil from the pyrolysis of PP alone was 46.83 KJ/kg. Therefore, the co-pyrolysis of biomass with plastic is an effective method to improve the characteristics of bio-oil fuels. When CS:PP = 1:1 was co-pyrolysis, the HHV of pyrolysis oil was 41.78 KJ/kg, which is comparable to the HHV of diesel (42–45 KJ/kg) and gasoline (42–46 KJ/kg) [29].

#### 3.1.4. NMR Analysis of Pyrolysis Oil

The chemical shift databases of various components in ^13^C and ^1^H NMR of pyrolysis oil are collected [21]. Table 4 and Table 5, respectively, show the contents of various functional groups and different types of protons in pyrolysis oils during pyrolysis at 500 °C with different proportions of CS/PP. Relatively more oxygen-containing functional groups were generated by the pyrolysis of CS alone. Compared with the pyrolysis of CS, there were only aliphatic carbon (94.21%) and aromatic carbon (5.79%) in the pyrolysis oils of PP, but no oxygen-containing functional groups. As PP was an aliphatic hydrocarbon polymer, the content of aromatic carbon in the pyrolysis oils was very small, which was consistent with the conclusions reached by Jin et al. [31] and Sharypov et al. [32] during the pyrolysis of PP. Therefore, when PP was added to CS for co-pyrolysis, the aliphatic carbon content in the pyrolysis oils was increased and oxygen-containing functional groups were reduced.

The experimental and theoretical contents of physicochemical properties in pyrolysis oils with different ratios of CS/PP co-pyrolysis at 500 °C were compared. The synergistic effects of carbon-containing functional groups and different types of proton are shown in Figure 2 and Figure 3, respectively. The contents of aromatic carbon (16.60%, 11.19%, 8.4%) were greater than the theoretical values (16.17%, 10.98%, 7.86%), which may be due to the Diels–Alder reaction between the olefins derived from PP and the furans derived from CS [33,34,35,36,37,38], promoting the formation of aromatic compounds. Due to the addition of PP, the effective hydrogen-carbon ratio (H/Ceff ratio) of the system was increased, which reduced generation of polycyclic aromatic hydrocarbons (PAHs), so the content of H-PAH was far less than the theoretical value. In addition, the contents of aliphatic C–O bonds and carbonyl groups in co-pyrolysis oils were less than the theoretical values, which also indicated that co-pyrolysis was indeed beneficial to the deoxidation of pyrolysis oil. It is worth noting that, in all aromatic carbons, the aromatic C–O bonds and aromatic C–C bonds were larger than the theoretical values, while aromatic C–H bonds were smaller than the theoretical values, which may be due to the recombination reaction of alkyl and hydroxyl radicals with benzene rings, thus replacing the hydrogen atoms on the benzene rings [39,40]. The content of H–PAH was less than the theoretical value, which further proved that the protons on the benzene ring were substituted. The aromatic methyl carbon content was greater than the theoretical value and the aromatic methyl proton content was less than the theoretical value, which further illustrated that alkyl substitution occurred on the benzene ring and the protons on the substituted alkyl carbon atoms decreased. The aromatic hydroxyl proton content was greater than the theoretical value, which further validated the combination of hydroxyl radicals with benzene rings. Therefore, the possible collaborative reaction pathways of CS/PP co-pyrolysis are shown in Figure 4.

### 3.2. Co-Pyrolysis Analysis of CS and PP at Different Temperatures

#### 3.2.1. Product Yield Analysis

The co-pyrolysis of CS:PP = 1:1 at different temperatures (450 °C, 500 °C, 550 °C, 600 °C, 650 °C) was performed, and the yields of gas, carbon and oil are shown in Figure 5. As shown in Figure 5, the yield of pyrolysis oil first increased and then decreased with the increase of temperature, and the optimal yield was 44.8% at 550 °C. This is because the increase of temperature will promote the secondary pyrolysis reaction, and the liquid yield would decrease with the further increase of temperature. Although the carbon yield decreased from 22.4% at 450 °C to 11.2% at 650 °C, it is worth noting that there was little significant change from 550 °C (12.0%) to 650 °C (11.2%). With the carbon yield almost unchanged, the gas yield increased sharply from 43.2% at 550 °C to 52.7% at 650 °C, which is due to the conversion of steam into smaller organic molecules or other non-condensable gaseous products by secondary cracking [36]. Considering the maximum yield of co-pyrolysis oil obtained at 550 °C, the following discussions of the impact of other conditions on co-pyrolysis are employed 550 °C.

#### 3.2.2. NMR Analysis of Pyrolysis Oil

In addition to the product yield, the reaction temperature also plays an important role in the distribution of chemical components in pyrolysis oils. The percentages of different carbon-containing functional groups of pyrolysis oil produced by CS:PP = 1:1 co-pyrolysis at different temperatures are shown in Figure 6. With the increase of temperature, the aromatic carbon first increased and then decreased, reaching the maximum content of 16.56% at 600 °C. The aromatic C–C bonds and aromatic C–H bonds have the same trends as the aromatic carbon with the temperature change, reaching the maximum values of 6.44% and 7.91% at 600 °C, respectively. The aromatic C–O bonds decreased with the increase of temperature and reached a minimum value of 1.10% at 650 °C. Different from aromatic carbon, the oxygen-containing functional groups decreased first and then increased with temperature, and reached the minimum content of 4.21% at 600 °C, which indicated that 600 °C was conducive to the aromatization and deoxidation reaction in the co-pyrolysis process. In addition, the aliphatic C–C bonds and the aromatic methyl groups first increased and then decreased with the increase of temperature, reaching the maximum contents of 84.70% and 28.25% at 500 °C and 550 °C, respectively. At high temperature, the linear hydrocarbons from PP form alkenes through end-chain β-breaking and hydrogen abstraction reactions, and then form benzene and its derivatives through a cyclization reaction, thus reducing the formation of aliphatic hydrocarbons [41,42,43]. 

### 3.3. Co-Pyrolysis Analysis of CS and PP with Different Catalyst Ratios

#### 3.3.1. Product Yield Analysis

The yields of co-pyrolysis at different feedstock / catalyst ratios (1:0, 1:2, 1:3, 1:4, 1:5) at 550 °C with CS:PP = 1:1 is shown in Figure 7. As shown in Figure 7, the addition of catalyst significantly affected the distribution of gas, carbon and oil during the co-pyrolysis of CS and PP. With the increase of catalyst amount, the carbon yield increased to 16.7%; the oil yield decreased to 22.5%. Further, the gas yield increased first and then decreased, and reached a maximum of 61.2% at feedstock:catalyst = 1:4. As the residence time of the pyrolysis steam through the catalytic micropores is prolonged with the increase of the catalyst, the steam generated by the co-pyrolysis of CS and PP undergoes secondary cracking through excessive HZSM-5 zeolite, which results in more condensable oils being converted into gas. In addition, a higher catalytic amount facilitates the generation of PAHs, mainly naphthalene [44], which are the precursors of coke, leading to an increase in carbon yield as the catalyst amount increases.

#### 3.3.2. NMR Analysis of Pyrolysis Oil

In addition to the product yield, the influence of catalyst on the chemical composition of pyrolysis oil is shown in Figure 8. When the ratio of catalyst to feedstock was lower than 3:1, HZSM-5 did not significantly improve the aromatic carbon in co-pyrolysis oils, which may be due to the presence of CS leading to catalyst deactivation, resulting in the reduction of the aromatic carbon conversion rate. With the further increase in the amount of catalyst, the aromatic C–C bonds and aromatic C–H bonds increased significantly, which may be the high catalyst amount promoted the Diels-Alder reaction during the co-pyrolysis process [33]. In addition, the addition of HZSM-5 promoted the deoxidation reaction in the co-pyrolysis process. When catalyst:feedstock = 2:1, the carbonyl content was 0. When catalyst:feedstock = 3:1, the content of levoglucosan was 0. When catalyst:feedstock = 5:1, the aliphatic C-O bonds and methoxyl content reached their minimum values of 0.44% and 0.03%, respectively. It is worth noting that the presence of HZSM-5 significantly reduced the content of aromatic methyl carbon in pyrolysis oils, promoting the dealkylation reaction during the co-pyrolysis process. 

### 3.4. Co-Pyrolysis Analysis of CS and PP in Different Atmospheres

#### 3.4.1. Product Yield Analysis

The co-pyrolysis of CS:PP = 1:1 under different atmospheres (N_2_, CO_2_) at 550 °C was conducted. The yields of gas, carbon and oil are shown in Figure 9. The production rates of oil, gas and carbon were 44.8%, 43.2% and 12.0% under N2 atmosphere, respectively. Obviously, the co-pyrolysis yield has obvious difference under different atmosphere. Compared with N_2_ atmosphere, the oil and carbon yields in CO_2_ atmosphere decreased by 5.6% and 0.4% respectively, while the gas yield increased by 6.0%. The slight reduction in carbon yield is due to the reaction of carbon with CO_2_, which is limited by temperature. In a CO_2_ atmosphere below 700 °C, there is almost no significant difference in the weight loss of carbon [23]. The decrease of oil yield is due to the CO_2_ atmosphere promoting the further decomposition of oil, which in turn forms small molecules of gaseous compounds, leading to the increase of gas yield in the pyrolysis system.

#### 3.4.2. NMR Analysis of Pyrolysis Oil

In addition to the product yield, the reaction atmosphere also significantly affects the distribution of chemical components in pyrolysis oils. The percentage of carbon-containing functional groups of oil produced by co-pyrolysis in different reaction atmospheres is shown in Figure 10. Compared with N_2_ atmosphere, the content of all oxygen-containing functional groups increased in the CO_2_ reaction atmosphere, which may be due to the participation of CO_2_ in the reaction of pyrolysis oil. In addition, the contents of Aromatic C–C bonds and Aromatic C–H bonds reduced to 2.41% and 4.65%, respectively, and the total aromatic content reduced by 2.1%. In summary, the influence of reaction atmosphere on the co-pyrolysis is obvious.

## 4. Conclusions

The effects of mixing ratio of CS and PP, pyrolysis temperature, addition amount of catalyst (HZSM-5) and reaction atmosphere (N_2_ and CO_2_) on the properties of pyrolysis products were studied. The study showed that due to the interaction between CS and PP, the co-pyrolysis not only had a promoting effect on the production of oil, but also influenced a great change in properties. Compared with CS pyrolysis alone, the co-pyrolysis oils had lower oxygen content and higher HHV, which are more conducive to their future application in industry. In addition, with the increase of pyrolysis temperature, 550 °C was beneficial for maximizing yield and 600 °C was favorable for aromatization and deoxygenation during co-pyrolysis. Although the addition of catalyst (HZSM-5) reduced the production of co-pyrolysis oil, it improved the quality of oil and promoted the generation of aromatic hydrocarbons. Compared to N_2_ atmosphere, CO_2_ had a certain oxidation effect on the components of pyrolysis oil, which reduced the production of co-pyrolysis oil and promoted the increase of oxygen-containing aromatics in co-pyrolysis oils. In terms of oil yield and degree of aromatization, the optimal co-pyrolysis condition of CS:PP = 1:1 was 550 °C under the N_2_ atmosphere with the catalyst addition of 1:4.

## Figures and Tables

**Figure 1 polymers-12-00973-f001:**
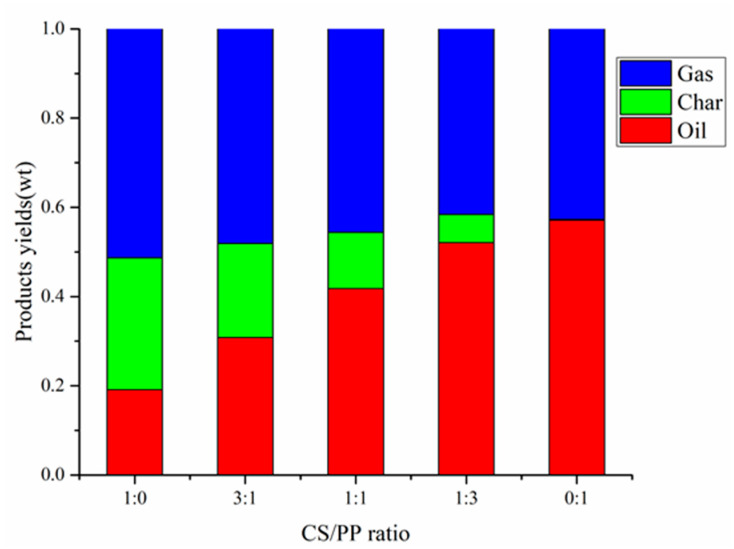
Products yields of different CS / PP ratios at 500 °C co-pyrolysis.

**Figure 2 polymers-12-00973-f002:**
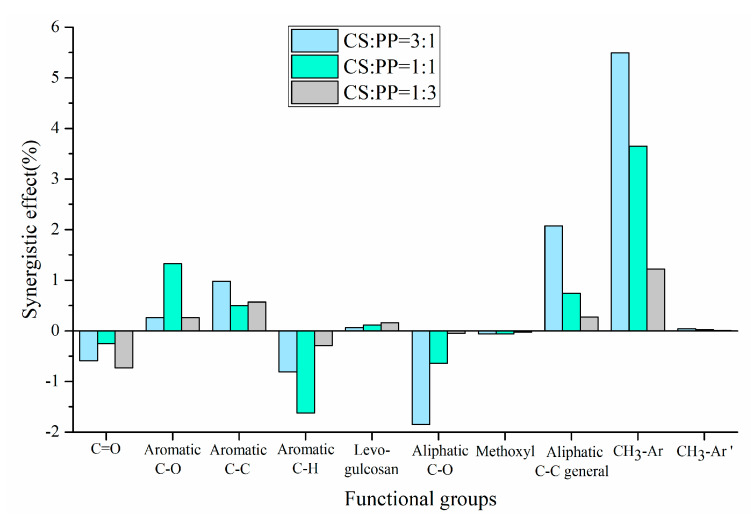
Synergistic effect of carbon-containing functional groups of pyrolysis oil produced from different CS/PP ratios at 500 °C co-pyrolysis.

**Figure 3 polymers-12-00973-f003:**
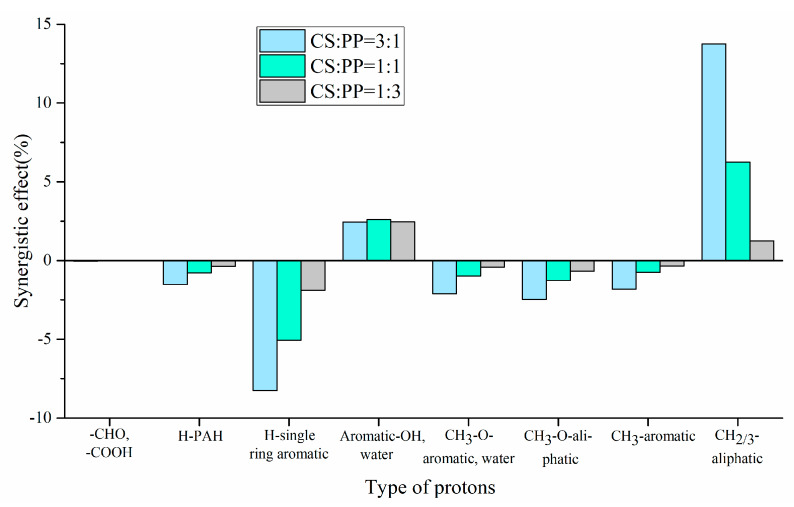
Synergistic effect of type of protons of pyrolysis oil produced from different CS/PP ratios at 500 °C co-pyrolysis.

**Figure 4 polymers-12-00973-f004:**
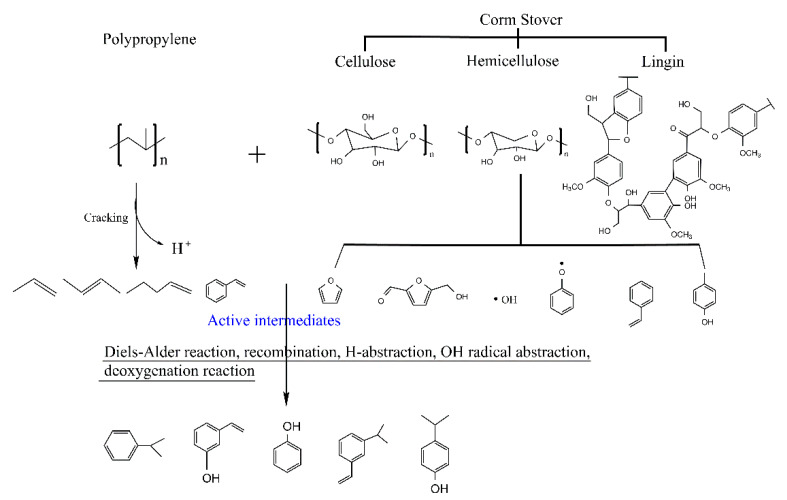
Possible synergistic reaction pathways in CS/PP co-pyrolysis.

**Figure 5 polymers-12-00973-f005:**
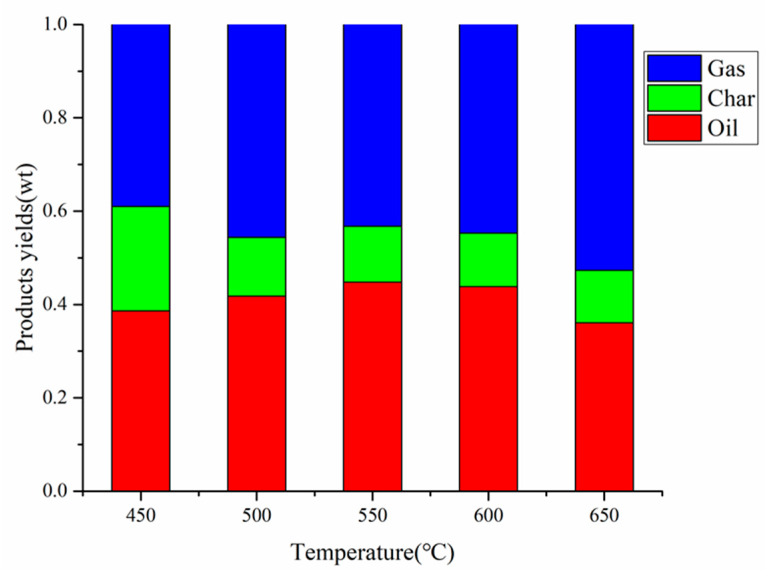
Products yields by CS:PP = 1:1 co-pyrolysis at different temperatures.

**Figure 6 polymers-12-00973-f006:**
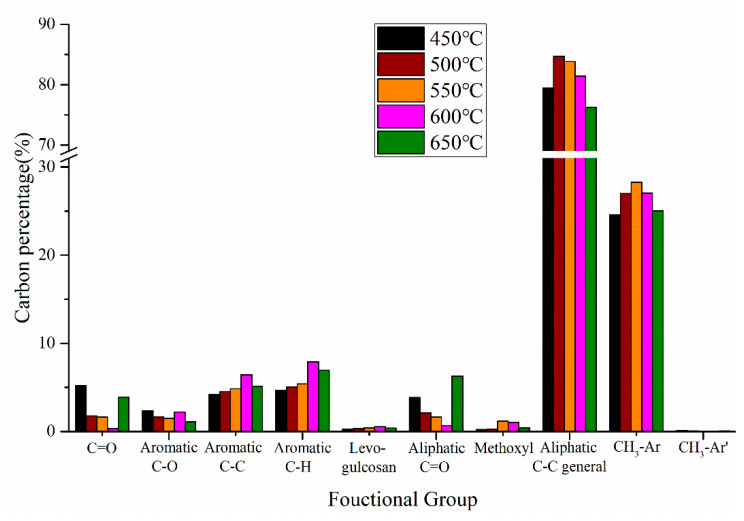
Carbon percentage of pyrolysis oil produced by CS:PP = 1:1 co-pyrolysis at different temperatures.

**Figure 7 polymers-12-00973-f007:**
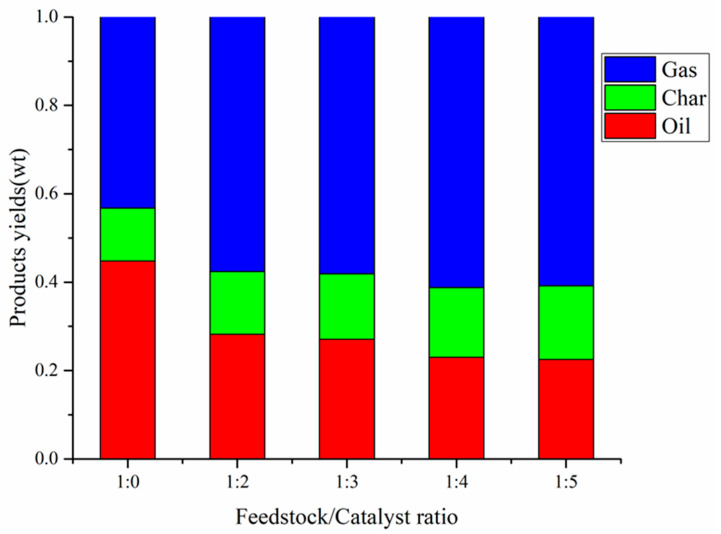
Products yields by CS:PP = 1:1 co-pyrolysis at 550 °C under different catalyst ratios.

**Figure 8 polymers-12-00973-f008:**
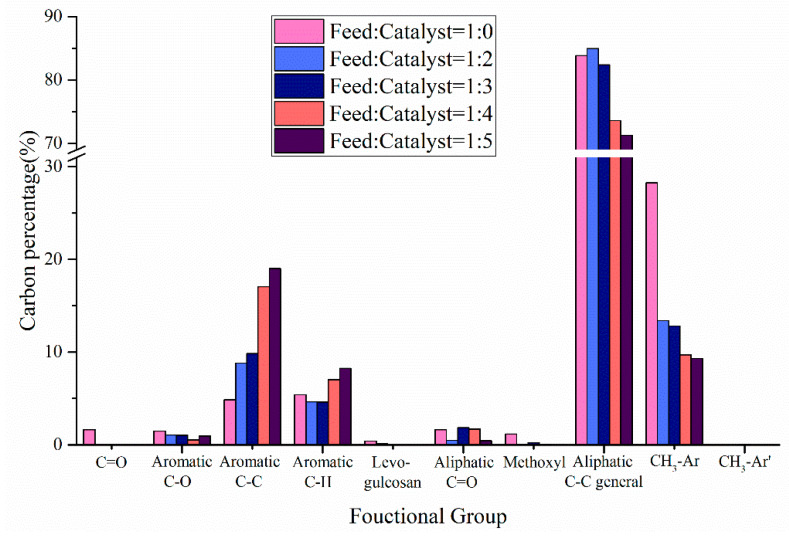
Carbon percentage of pyrolysis oil produced by CS:PP = 1:1 co-pyrolysis at 550 °C under different catalyst ratios.

**Figure 9 polymers-12-00973-f009:**
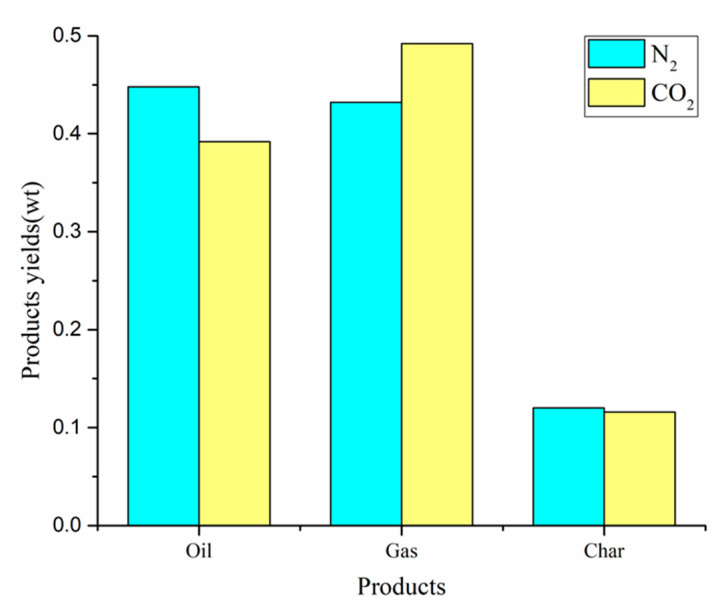
Products yields by CS:PP = 1:1 co-pyrolysis at 550 °C under different atmospheres.

**Figure 10 polymers-12-00973-f010:**
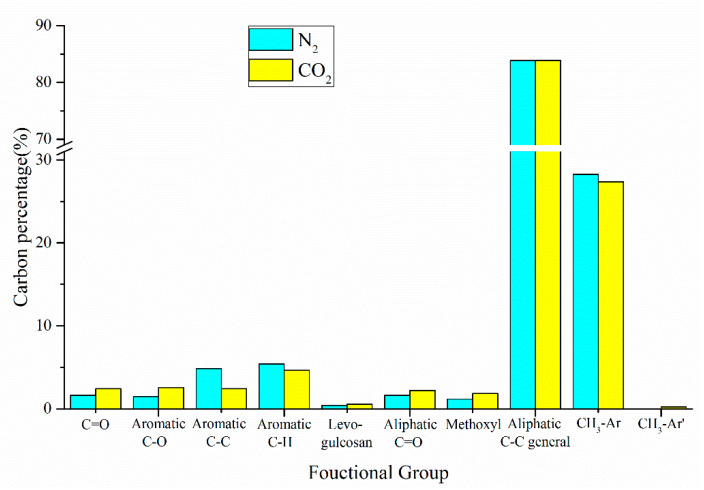
Carbon percentage of pyrolysis oil produced by CS:PP = 1:1 co-pyrolysis at 550 °C under different atmospheres.

**Table 1 polymers-12-00973-t001:** Proximate and ultimate analysis of CS and PP.

Sample		Proximate Analysis *w_d_*/%			Ultimate Analysis *w_d_*/%			
	Moisture	Ash	Volatile	Fixed Carbon	C	H	O *	N
CS	2.73	6.43	74.06	16.78	42.93	6.38	49.5	1.19
PP	0	0.06	99.82	0.12	85.43	14.57	0	0

* by difference.

**Table 2 polymers-12-00973-t002:** Experimental and calculated yields of different CS / PP ratios at 500 °C co-pyrolysis.

Sample	Experimental Yield			Calculated Yield			Synergistic Effect		
CS:PP	Oil (%)	Char (%)	Gas (%)	Oil (%)	Char (%)	Gas (%)	Oil (%)	Char (%)	Gas (%)
1:0	19.1	29.6	51.3	-	-	-	-	-	-
3:1	30.8	21.1	48.1	28.6	22.25	49.15	2.2	−1.15	−1.05
1:1	41.8	12.6	45.6	38.1	14.9	47	3.7	−2.3	−1.4
1:3	52.1	6.3	41.6	47.6	7.55	44.85	4.5	−1.25	−3.25
0:1	57.1	0.2	42.7	-	-	-	-	-	-

**Table 3 polymers-12-00973-t003:** C, H and O analysis of pyrolysis oil produced from different CS / PP ratios at 500 °C co-pyrolysis.

Sample	Pyrolysis Oil Composition (wt %)			Deoxygenation ^1^ (%)	HHV (KJ/kg)
CS:PP	C	H	O		
1:0	63.0	7.4	29.6	40.9	28.44
3:1	76.3	10.9	12.8	65.9	38.60
1:1	82.4	11.3	6.3	74.9	41.78
1:3	84.4	13.9	1.7	86.4	45.84
0:1	85.8	14.2	0	-	46.83

^1^ Deoxygenation = (O_feed_ − O_oil_)/O_feed_ [30].

**Table 4 polymers-12-00973-t004:** Carbon percentage of pyrolysis oil produced from different CS / PP ratios at 500 °C co-pyrolysis.

Functional Group	Integration Region (ppm)	CS:PP = 1:0	CS:PP = 3:1	CS:PP = 1:1	CS:PP = 1:3	CS:PP = 0:1
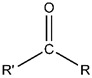	215.0–166.5	8.00	3.42	1.76	0.07	0
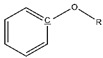	166.5–142.0	1.19	0.86	1.63	0.38	0
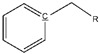	142.0–125.0	9.65	6.88	4.53	3.48	2.17
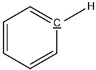	125.0–95.8	15.69	8.86	5.03	4.54	3.62
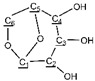	C_1_ 102.3, C_2_ 72.0, C_3_ 73.7, C_4_ 71.7, C_5_ 76.5, C_6_ 64.9	0.89	0.51	0.33	0.25	0
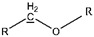	95.8–60.8	10.92	3.62	2.10	1.05	0
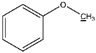	60.8–55.2	1.25	0.57	0.25	0.10	0
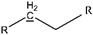	55.2–0.0	53.30	75.79	84.70	90.38	94.21
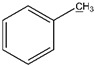	21.6–19.1	8.64	23.92	26.99	27.51	28.26
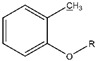	16.1–15.4	0.15	0.12	0.06	0.03	0

**Table 5 polymers-12-00973-t005:** Hydrogen percentage of pyrolysis oil produced from different CS / PP ratios at 500 °C co-pyrolysis.

Type of Protons	Ranges (ppm)	Hydrogen Percentages				
		CS:PP = 1:0	CS:PP = 3:1	CS:PP = 1:1	CS:PP = 1:3	CS:PP = 0:1
**–CHO, –COOH**	9.6–10.0	0.16	0.05	0.03	0.01	0
**H-PAH**	7.5–9.0	3.90	0.48	0.26	0.10	0.09
**H–single ring aromatic**	6.0–7.5	21.84	2.90	0.74	0.69	0.43
**Aromatic–OH, water**	~4.0–5.0	6.21	5.55	4.17	3.07	0
**CH_3_–O–aromatic, water**	~3.8	5.17	0.48	0.32	0.11	0
**CH_3_–O–aliphatic**	~3.3	11.46	3.27	1.61	0.48	0
**CH_3_–aromatic**	~2.2	7.35	2.67	2.29	1.83	1.60
**CH_2/3_–aliphatic**	0.0–2.0	43.92	84.60	90.58	93.71	97.88

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
