# Peer review of "Effects of Different Conditions on Co-Pyrolysis Behavior of Corn Stover and Polypropylene"

_polymers, 2020, doi:10.3390/polym12040973_

Round 1
Reviewer 1 Report
polymers-768871
The authors present in their work entitled ‘Effects of different conditions on co-pyrolysis behavior of corn stover and polypropylene’ a screening of parameters to arrive at optimised conditions for the title valorisation approach.
The work is structured well as such, and processes generating the various data discussed are by and large clearly presented. The analysis of the oils by NMR represents an unconventional element of this study, whose principal argument is not new as such. Some aspects need nevertheless attention before eventually publishing this work.
Specific comments:
- The paper is too long; discussions of the various screenings of parameters are repetitive as such. The paper would benefit a lot from a detailed, quasi exemplary discussion as it is done in chapter 3.1, with a summary of the results obtained in the other screenings, currently discussed extensively in chapters 3.2, 3.3 and 3.4. Main results of chapters 3.2, 3.3 and 3.4 should be summarised by combining Figures 5 and 6, and combining Tables 6, 7 and 8, omitting to list anew the integration ranges. Numbers should be adjusted to contain only significant figures, and errors for the NMR-based analysis should be estimated.
- Regarding the NMR analysis of the oils: the argument brought forward by the authors for the use of NMR for analysis to omit shortcomings of a gaschromatographic analysis can be understood from some points of view. However, missing the separation aspect, NMR is known to suffer from peak overlays, etc. With respect to the fingerprinting aspect needed in the context of this work, the use of two-dimensional HSQC might allow for a more detailed picture. Has this been tried/evaluated? NMR is further known to be rather insensitive to trace components, especially in complex compound mixtures. Was the NMR fingerprinting used in here benchmarked using established/accepted GC-analysis? The paper would benefit from such a direct comparison for one or two samples.
- It might be beneficial in general to represent all the NMR results in a single graphic representation, such as to allow for a rather facile detection of trends for the formation of certain species as function of the chosen conditions.
- Table 4 is of low quality, and evokes a bit the impression of being reproduced. A better quality is needed for this figure; writing across the ‘reaction error’ should be avoided.
- The authors should clearly and explicitly state the conditions identified as optimum considering all tested options.
- The English needs another round of sound proofreading, eliminating some inaccuracies like (‘deuterium chloroform’).
Given that the authors consider/answer the aforementioned points during a major revision, the work should become suitable for publishing in polymers.
Author Response
Dear Reviewer,
Thank you very much for your valuable comments on our manuscript. We have revised and answered your question in the attachment, please refer to it.
Kind regards,
Fengze Wu

Reviewer 2 Report
The article evaluates the pyrolysis of corn stover and polypropylene. The influence of mixing ratio, temperature, amount of catalyst and reaction atmosphere conditions were studied.
The corresponding comments are listed below:
- In my opinion the abstract is well written and allows the reader to get a preliminary overview of the article's content.
- In general terms, the introduction is correctly written and shows a good review of the topic of the article. However, a further emphasis on the novelty of the article is lacked. It would also be interesting for the authors to provide data. For example, on line 49, what increments and in what values? Or in line 39-40, numerically, which has represented the co-pyrolysis technology in terms of the amount of pyrolysis oil.
- Materials and methods.
- Extensive work has been done and a flowchart should be added to help the lector to follow the experimental design.
- The names, models, references and main characteristics of the equipment used should be added.
- Results and discussion.
- Why the authors didn’t consider the ratios 1:2 and 2:1 of CS/PP?
- Figure 2 doesn’t have enough quality, the x axis it’s difficult to read.
- Figure 3 doesn’t have enough quality, the x axis it’s difficult to read.
- The results and discussion section it’s well written and the results are relevant.
- The conclusions are adequate and supported by the obtained results.
Author Response
Dear Reviewer,
Thank you very much for your high opinion and valuable comments on our manuscript. We have modified the question you raised in the attachment, please refer to it.
Kind regards,
Fengze Wu

Round 2
Reviewer 1 Report
The authors presented an adequately revised manuscript, that is now suitable for publication; the English would still benefit from anotehr check by authors and eventually editors.